# New Advances in the Research of Resistance to Neoadjuvant Chemotherapy in Breast Cancer

**DOI:** 10.3390/ijms22179644

**Published:** 2021-09-06

**Authors:** Junsha An, Cheng Peng, Hailin Tang, Xiuxiu Liu, Fu Peng

**Affiliations:** 1Key Laboratory of Drug-Targeting and Drug Delivery System of the Education Ministry, Sichuan Engineering Laboratory for Plant-Sourced Drug and Sichuan Research Center for Drug Precision Industrial Technology, Department of Pharmacology, West China School of Pharmacy, Sichuan University, No. 17, Section 3, Southern Renmin Road, Wuhou District, Chengdu 610041, China; 2017141493006@stu.scu.edu.cn (J.A.); liuxiuxiu@scu.edu.cn (X.L.); 2State Key Laboratory of Southwestern Chinese Medicine Resources, Chengdu University of Traditional Chinese Medicine, Chengdu 610075, China; pengcheng@cdutcm.edu.cn; 3Department of Breast Oncology, Sun Yat-sen University Cancer Center, Guangzhou 510275, China; tanghl@sysucc.org.cn

**Keywords:** neoadjuvant chemotherapy, breast cancer, drug resistance

## Abstract

Breast cancer has an extremely high incidence in women, and its morbidity and mortality rank first among female tumors. With the increasing development of medicine today, the clinical application of neoadjuvant chemotherapy has brought new hope to the treatment of breast cancer. Although the efficacy of neoadjuvant chemotherapy has been confirmed, drug resistance is one of the main reasons for its treatment failure, contributing to the difficulty in the treatment of breast cancer. This article focuses on multiple mechanisms of action and expounds a series of recent research advances that mediate drug resistance in breast cancer cells. Drug metabolizing enzymes can mediate a catalytic reaction to inactivate chemotherapeutic drugs and develop drug resistance. The drug efflux system can reduce the drug concentration in breast cancer cells. The combination of glutathione detoxification system and platinum drugs can cause breast cancer cells to be insensitive to drugs. Changes in drug targets have led to poorer efficacy of HER2 receptor inhibitors. Moreover, autophagy, epithelial–mesenchymal transition, and tumor microenvironment can all contribute to the development of resistance in breast cancer cells. Based on the relevant research on the existing drug resistance mechanism, the current treatment plan for reversing the resistance of breast cancer to neoadjuvant chemotherapy is explored, and the potential drug targets are analyzed, aiming to provide a new idea and strategy to reverse the resistance of neoadjuvant chemotherapy drugs in breast cancer.

## 1. Background

Breast cancer is currently a cancer with an extremely high incidence in women, and its mortality rank first among female tumors [1]. According to the data of GLOBOCAN in 2018, about 2.1 million patients are diagnosed with breast cancer, and the death toll is 630,000 [2]. Today, with changes in the environment and lifestyle, the incidence of breast cancer is also increasing [3]. New statistics show that breast cancer still ranks first among female cancers. Therefore, overcoming breast cancer has increasingly become a problem of global concern.

In clinical treatment of breast cancer, surgery is usually combined with chemotherapy. With the development of biology and immunology, the approach to breast cancer treatment is constantly updated. In recent years, breast cancer has been considered a systemic disease, and neoadjuvant chemotherapy has also been included as an important part of the treatment of breast cancer.

Neoadjuvant chemotherapy refers to systemic chemotherapy before the implementation of local treatment methods (such as surgery or radiotherapy). It is mainly suitable for patients with mid-stage and locally advanced breast cancer. The concept of neoadjuvant chemotherapy was formally proposed by Rosen et al. in 1979. It aims to transform inoperable breast cancer into operable breast cancer, convert breast cancer that requires breast removal into breast-sparing breast cancer and provide drug basis in the follow-up treatment to improve the prognosis of patients [4].

At present, there is no uniform standard for neoadjuvant chemotherapy for breast cancer. In the early stage, a unified neoadjuvant chemotherapy regimen is generally given to all patients, but today’s neoadjuvant chemotherapy regimens tend to be personalized, as shown in Figure 1, generally based on curative effect prediction markers and molecular subtypes to give personalized treatment.

In the use of chemotherapy drugs, anthracyclines, such as doxorubicin and epirubicin, are generally used in combination with drugs such as cyclophosphamide, epirubicin, and fluorouracil. The emergence of taxanes and their significant anti-tumor activity against advanced breast cancer have further improved the efficacy of neoadjuvant chemotherapy. In clinical use, no matter single drug or combination drugs, taxane drugs all show good anti-tumor activity. For breast cancer subtypes with HER2 overexpression, trastuzumab is generally added to the neoadjuvant chemotherapy regimen, and satisfactory results have been obtained in clinical use [5].

Although the efficacy of neoadjuvant chemotherapy has been confirmed, clinical trial data show that the effects of neoadjuvant chemotherapy for different breast cancer patients are very different, and it is easy to develop drug resistance, which is not conducive to subsequent treatment [6]. Drug resistance to neoadjuvant chemotherapy is one of the main reasons for its treatment failure, and it is one of the most challenging problems in the treatment of breast cancer today. Therefore, this article will focus on the drug resistance mechanism of neoadjuvant chemotherapy and discuss the follow-up solutions to provide references for the clinical application of neoadjuvant chemotherapy.

## 2. Drug Resistance Mechanism of Neoadjuvant Chemotherapy in Breast Cancer

### 2.1. Drug Metabolizing Enzymes

Breast cancer cells have a unique metabolic pattern during the growth process. This pattern causes breast cancer cells to produce an abnormal internal environment, which makes the cells’ endogenous metabolism change tremendously, as well as interact with exogenous cells. The expression and activity of drug metabolizing enzymes (DMEs) related to cancer cell metabolism are also different from normal cells [7]. The design of many antitumor drugs requires metabolic activation in tumor cells to obtain clinical efficacy, and neoadjuvant chemotherapy drugs for breast cancer are no exception. However, breast cancer cells can also develop drug resistance by reducing drug activation or metabolism to inactivate compounds.

It is generally believed that the metabolic process of drugs is closely related to the activity of cytochrome P450 (CYP). The CYP1, CYP2 and CYP3 families play a vital role in the metabolism of anti-tumor drugs [8,9]. CYP-mediated catalytic reactions can convert drug molecules into polar hydrophilic metabolites, which are further eliminated from the body. For some oral neoadjuvant chemotherapy drugs for breast cancer, exposure to CYP3A after administration is almost inevitable, and the catalytic reaction mediated by CYP3A strongly limits the oral bioavailability of the drug, which usually leads to drug inactivation [10,11]. Related studies have shown that the expression of CYP3A4 is negatively correlated with the sensitivity of breast cancer patients to paclitaxel drugs in neoadjuvant chemotherapy, which may be related to changes in bioavailability caused by the interaction of CYP3A and P-gp [12].

The CYP-mediated drug metabolism of breast cancer tissues is also closely related to the development of breast cancer drug resistance. Iscan et al. [13] studied the expression of CYPs in breast cancer tissues, and the results showed that in comparison with normal tissues, the expression of CYP1A1 is lower, and CYP2A6, 2F1, 2A7, 2A13, 3A4, 3A5, and 3A7 are not expressed. El-Rayes et al. [14] reported that the high expression of GST-Pi in breast cancer tissues is closely related to treatment resistance. Martinez et al. [15] compared the expression of CYP450 superfamily members and other transcription factors related to drug metabolism by analyzing control samples of breast cancer and normal breast tissues, and the results showed that the expression levels of CYP1B1, CYP2A6, GST-A4, etc., are increased in breast cancer. Continuous research has shown that CYP1B1 is related to tumor drug resistance, laying the foundation for CYP1B1 as a new target for reversing breast cancer neoadjuvant chemotherapy drugs resistance [16,17]. In addition, according to related literature, CYP 2C9 is overexpressed in vascular endothelial cells inducing cell migration and angiogenesis and is also a potential target [18].

The expression of DMEs is closely related to the metabolism of neoadjuvant chemotherapy drugs in breast cancer cells. At the same time, it may also cause cell resistance and affect the therapeutic effect. Further study of the relationship between its expression and neoadjuvant chemotherapy resistance has important clinical significance and is required to improve patient treatment.

### 2.2. Drug Efflux System

The concentration of drugs in breast cancer cells is closely related to transmembrane proteins. Neoadjuvant chemotherapy drugs can be transported out of breast cancer cells through transmembrane proteins, thereby reducing drug concentration and leading to cell resistance. The mechanism that mediates drug efflux is mainly the ATP-binding cassette (ABC) transporter superfamily. Increased expression levels or enhanced functions can mediate tumor multidrug resistance (MDR), which is also one of the most studied drug resistance mechanisms [19].

P-glycoprotein (P-gp) is one of the drug efflux pumps of the ABC transporter family, and it is the first transporter to be identified and widely studied [20]. The main mechanism of P-gp induced drug resistance is associated with the use of energy released by ATP hydrolysis to pump the neoadjuvant chemotherapy drugs out of the cell, so that the drug concentration is lower than the effective concentration [21]. Relevant studies have shown that chromosome 7q11.2–21 is amplified and the ABCB1 gene encoding P-gp is fused with the transcription of the upstream gene SLC25A40, which leads to the overexpression of P-gp and causes breast cancer cells to develop drug resistance [22].

Most studies still focus on reversing neoadjuvant chemotherapy resistance of breast cancer by inhibiting the expression of P-gp protein, but a new study recently provides a new idea for reversing breast cancer resistance [23]. The ATP-dependent transporter P-gp has a large energy demand in the process of functioning, and this study introduces the P-gp substrate, namely verapamil, into drug-resistant cells, which significantly reduces the intracellular ATP level, thereby causing ATP consumption. Finally, the continuous peak activity of cell oxidative phosphorylation produces reactive oxygen species, which leads to cytotoxicity and reverses breast cancer resistance to neoadjuvant chemotherapy.

In addition, multi-drug resistance protein (MRP) and breast cancer resistance protein (BCRP), which belong to the same ABC transporter family as P-gp, can also function as a “drug pump”. When neoadjuvant chemotherapy drugs are used for a long time, the genes encoding the above-mentioned proteins are overexpressed, which increases drug efflux and causes resistance to neoadjuvant chemotherapy [24,25].

Many different types of breast cancer studies have shown that increased expression of any transporter can lead to poor clinical results. Various transporter inhibitor drugs have been used clinically to minimize drug efflux. There are currently studies on the use of traditional Chinese medicine to reverse tumor multidrug resistance. Although it has not been applied to breast cancer, it can provide a reference for the subsequent neoadjuvant chemotherapy resistance of breast cancer [26,27].

### 2.3. Glutathione Detoxification System

Glutathione (GSH) detoxification system is composed of glutathione and a series of related enzymes. It has the function of eliminating oxygen free radicals in the cell, maintaining the normal form of protein and maintaining the normal oxidation-reduction environment. It is an important system for cells to resist carcinogenesis and damage [28]. Glutathione-S-transferase (GST) is one of the related enzymes of the glutathione detoxification system. In this detoxification system, GSTs have been studied most thoroughly. Overexpression of GSTs is closely related to tumor resistance [29]. Recent experiments have shown that GSH can have a synergistic effect with multidrug resistance-associated protein (MRP) to eliminate neoadjuvant chemotherapy drugs from breast cancer cells and develop drug resistance [30,31].

Platinum drugs are one of the neoadjuvant chemotherapeutic drugs for breast cancer, and the resistance of breast cancer to platinum drugs is closely related to the GSH detoxification system. After entering the cell, platinum drugs complex with GSH and are retained in the cytoplasm and cannot enter the nucleus [32,33]. Therefore, reducing the level of GSH in breast cancer cells can be an effective means to reverse tumor drug resistance [34]. Relevant studies have shown that the inorganic material MnO_2_ can undergo redox reactions in a high GSH environment, which can reduce the level of intracellular GSH, thereby reducing its combination with platinum drugs and achieving the purpose of reversing the drug resistance of breast cancer cells [35,36]. Therefore, currently carrier materials modified with MnO_2_ to target the delivery of neoadjuvant chemotherapy drugs to breast cancer cells are being considered for the reversal of drug resistance of breast cancer cells mediated by the GSH detoxification system.

### 2.4. Changes in Drug Targets

The efficacy of drugs is affected by changes in drug targets. In breast cancer cells, changes in drug-related targets can lead to drug resistance. In particular, most neoadjuvant therapies with targeted therapy as an idea will develop drug resistance during the application process.

There are two types of targeted drugs used in neoadjuvant chemotherapy for breast cancer, including HER2 receptor inhibitors. HER2 belongs to the family of receptor protein tyrosine kinases (RPTKs) and is overexpressed in about 20% of breast cancer patients [37]. The targeted preparations developed for HER2 are not only used in neoadjuvant therapy, but also in the normal treatment of breast cancer [38]. Trastuzumab is the first HER2 targeted drug, and its main mechanism is to bind with specific domains of the HER2 receptor to inhibit signaling mediated by HER2. However, clinically, the problem of trastuzumab primary or secondary drug resistance has become increasingly prominent. Russo et al. [39] believe that this resistance is related to nuclear HER2 translocation, while Mohd-Nafi et al. [40] found that long-term use of trastuzumab induces nuclear HER4 up-regulation to obtain drug resistance.

The overexpression of p95HER2 and the overexpression of cell surface mucin 4 (MUC4) on breast cancer cells may affect the HER2 target and hinder the binding of trastuzumab to HER2, which is related to poor drug efficacy [41,42].

The development of other targeted preparations for different targets has greatly improved the problems caused by single drug target resistance. Baselga et al. [43] combined trastuzumab and pertuzumab with two targeted preparations. The results showed that in comparison with the single-targeted therapy group, the dual-targeted therapy group had significantly better efficacy. For patients with single-targeted drug resistance in neoadjuvant chemotherapy, other target drug combinations can be considered to improve drug resistance.

### 2.5. DNA Damage Repair

DNA damage response (DDR) is a signal cascade reaction. When DNA is damaged, it can trigger the sensor molecular system and transmit the signal to the upstream sensor to cause DNA damage repair [44]. For neoadjuvant chemotherapy drugs that directly or indirectly damage DNA, the DNA damage repair mechanism can reverse the damage caused by the drug, resulting in drug resistance.

DDR is regulated by multiple genes, among which TDP1 is present in breast cancer cells and plays an important role in repairing DNA damage [45,46]. Wang et al. [47] found that overexpression of MIR-211 can act on the TDP1 target in breast cancer cells, thereby inhibiting its expression, further inhibiting DDR, and reversing the resistance of breast cancer to neoadjuvant chemotherapy drugs.

In addition, the DDR pathway is closely related to PARP-1 mediated nucleotide excision repair (NER) [48], and PARP-1 inhibitors have been used in the treatment of breast cancer patients in clinical applications [49]; therefore, when using neoadjuvant chemotherapy drugs that directly or indirectly target DNA damage leading to the development of drug resistance, PARP-1 inhibitors can be considered in combination therapy to enhance drug efficacy.

Recent new research shows that HORMAD1 can enhance the tolerance of DDR and promote the resistance of triple-negative breast cancer to chemotherapeutic drugs [50]. HORMAD1 is a specific germ cell protein that plays an important role in homologous chromosome recombination, and is reactivated in breast cancer, showing abnormally high expression [51]. Studies have shown that under the conditions of oxidative stress induced by related chemotherapy drugs, HORMAD1 may regulate the expression of enzymes in breast cancer cells to enhance the ability to resist apoptosis, which may be the main factor in inducing chemotherapy resistance.

Another type of neoadjuvant chemotherapeutic drug inducing DNA damage repair mediated drug resistance are the topoisomerase inhibitors. There are two main types of topoisomerases: Topo Ⅰ and Topo Ⅱ, which play a key role in DNA replication [52]. Topoisomerase inhibitors have significant effects in clinical applications, but drugs that target a single topoisomerase are prone to drug resistance. Studies have shown that its mechanism of inducing drug resistance is related to the overexpression of another DNA topoisomerase [53]. More and more clinical applications of topoisomerase dual inhibitors have effectively solved the drug resistance problem caused by a single topoisomerase. In the neoadjuvant chemotherapy of breast cancer, dual topoisomerase inhibitors such as flavonoids myricetin and fisetin can be selected to improve the drug resistance of patients [54].

### 2.6. Inhibition of Cell Apoptosis and Autophagy

This section may be divided by subheadings. It should provide a concise and precise description of the experimental results, their interpretation, as well as the experimental conclusions that can be drawn.

Apoptosis is a form of programmed cell death, which can occur through various mechanisms mediated by cellular mitochondria, endoplasmic reticulum stress, death receptors and other pathways [55]. Neoadjuvant chemotherapeutic drugs exert their pharmacological effects by promoting breast cancer cell apoptosis. At the same time, the resistance of tumor cells to apoptosis is one of the important mechanisms that cause breast cancer neoadjuvant chemotherapy resistance. 

Studies have found that overexpression of apoptosis-related factors, such as nuclear transcription factor (NF-κB) and B-cell lymphoma 2 (Bcl-2), can inhibit cell apoptosis, thereby reducing the sensitivity of breast cancer cells to drugs and causing drug resistance [56]. Reports on another drug resistance mechanism show that overexpression of the anti-apoptotic protein c-Flip can prevent the activation of pro-caspase-8, thereby inhibiting breast cancer cells apoptosis and producing drug resistance [57,58,59].

Autophagy is called type II programmed cell death, which is a process in which cell lysosomes engulf damaged organelles and cellular macromolecular for degradation and reuse [60]. Among the various signal pathways involved in autophagy, ULK1, P13 Kinase Class Ⅲ, AMPK promote autophagy, while mTOR inhibits autophagy by inhibiting ULK1. At the same time, PI3K/Akt Signaling and MAPK/Erk1/2 Signaling promote mTOR, while p53/Genotoxic Stress and AMPK inhibit the mTOR pathway [61,62].

In most studies, autophagy and apoptosis are antagonistic. Autophagy does not cause cell death, but promotes cell survival [63]. Relevant studies have shown that many neoadjuvant chemotherapy drugs can induce autophagy in breast cancer cells. However, autophagy under this condition is often a protective mechanism for breast cancer cells to resist apoptosis caused by neoadjuvant chemotherapy drugs, which is often not conducive to treatment [64].

Autophagy caused by endoplasmic reticulum stress is an important mechanism for the development of drug resistance. Endoplasmic reticulum stress is the state of cellular stress caused by the administration of neoadjuvant chemotherapy drugs, which can affect protein processing in breast cancer cells and disrupt the proper folding of protein precursors into functional proteins, and these misfolded proteins accumulate in the lumen of the endoplasmic reticulum [65]. In order to alleviate this stress state, breast cancer cells induce autophagy to clear proteins and cause drug resistance [66]. In addition, neoadjuvant chemotherapy drugs cause DNA damage, and autophagy can be induced after DNA damage [67]. In this case, autophagy can reduce cell apoptosis caused by DNA damage, which is not conducive to the treatment of breast cancer [68].

Neoadjuvant chemotherapy treats diseases by promoting the apoptosis of breast cancer cells, and the factors that can inhibit cell apoptosis can render breast cancer cells resistant. In recent years, continuous drug research targeting the autophagy pathway has shown that inhibiting the autophagy protection of breast cancer cells can abrogate the resistance to chemotherapeutics [69]. Besides, some autophagy inhibitors, such as 3-MA, Bafilomycin A1 and chloroquine, are used in combination with neoadjuvant chemotherapy drugs, which can effectively alleviate the drug resistance of breast cancer cells [70].

### 2.7. Epithelial-to-Mesenchymal Transition

Greenburg et al. [71] first introduced the concept of epithelial-to-mesenchymal transition (EMT), which refers to the loss of cell polarity and intercellular adhesion of polar epithelial cells, transforming into intercellular cells with migration ability [72]. There are three main types of EMT, of which type Ⅲ is closely related to the occurrence and development of tumors [73]. EMT plays an important role in breast cancer cell invasion and migration [74], and it also closely related to breast cancer neoadjuvant chemotherapy resistance.

Related studies have found that breast cancer cells that develop EMT are often resistant to neoadjuvant chemotherapy drugs, and the mechanism is related to the inhibition of cell apoptosis [75]. During the EMT process, the overexpression of interstitial phenotypic molecules such as N-cadherin and vimentin can lead to a decrease in intercellular adhesion, which is related to the strong migration and invasive ability of breast cancer cells, associated with drug resistance. One of the mechanisms is to up-regulate the expression of ABC transporter, thereby increasing the efflux of chemotherapeutic drugs reducing the efficacy and inducing drug resistance [76].

Various signaling pathways related to EMT have a significant impact on neoadjuvant chemotherapy drug resistance. The MAPK/ERK pathway is currently known as a pathway that can induce EMT, which is also related to the resistance of platinum drugs, one of the breast cancer neoadjuvant chemotherapeutic drugs [77,78]. Many studies have confirmed that the activation of the EGFR pathway is also related to the resistance of neoadjuvant chemotherapy drugs [79]. The TGFβ-SMAD pathway is the main driving force of EMT. For example, BMP9 and SMAD3 can both mediate EMT to make breast cancer cells resistant to neoadjuvant chemotherapy drugs [80,81]. Finally, the PI3K/AKT/NF-κB and JAK/STAT pathways are activated in breast cancer cells, and the activation of these two signaling pathways can produce neoadjuvant chemotherapy drug resistance, which is not conducive to patient treatment [82,83].

At present, there have been a large number of reports in the literature that EMT is one of the mechanisms of neoadjuvant chemotherapy resistance in breast cancer. The development of many inhibitors such as CDK4/6 inhibitors can cause abnormal expression of EMT-related genes in breast cancer cells, thereby inhibiting EMT and reducing the development of drug resistance [84]. In the follow-up treatment of breast cancer neoadjuvant chemotherapy, EMT-related inhibitors can be used in combination, which can solve the problem of poor efficacy caused by EMT-mediated drug resistance and improve the prognosis of patients.

### 2.8. Tumor Microenvironment

Tumor microenvironment (TME) is a complex integrated system. It is composed of tumor cells and extracellular matrix, a variety of stromal cells, immune cells, cytokines, etc. [85,86]. Due to their special metabolic state, tumor cells often have hypoxic and acidic environment and special structural components. Many studies have shown that TME plays an important role in breast cancer cells resistance [87,88].

Due to the rapid growth of breast cancer cells and the high level of metabolism, they are often relatively deficient in oxygen supply, so the cells are frequently in a state of hypoxia [89]. The hypoxic state of TME can induce drug resistance. Drug resistance to neoadjuvant chemotherapy in breast cancer is closely related to the role of cancer stem cells (CSCs), and studies have shown that the stem cell characteristics of CSC often depend on hypoxia and hypoxia-inducible factor (HIF) [90].

HIFs can participate in the resistance of breast cancer neoadjuvant chemotherapy through a variety of molecular pathways. In the TME, HIF-1α tends to be overexpressed, which can promote the increase in the expression of some drug-resistant proteins, and thus develop resistance to neoadjuvant chemotherapy drugs [91]. The expression of HIF-2α can promote conversion to a stem cell phenotype and enable breast cancer cells to acquire drug resistance by activating Wnt and Notch pathways [92]. In addition to regulating various signaling pathways to induce drug resistance, hypoxia can also induce autophagy to promote drug resistance [93].

Due to long-term hypoxia, the TME is often associated with an acidic environment. Compared with normal cells, the pH of breast cancer cells decreases [94]. Most of the currently used neoadjuvant chemotherapy drugs are weakly basic drugs, and their dissociation degree in body fluids is closely related to the environmental pH. In the acidic environment of TME, the proportion of dissociation of weakly basic drugs increases, and it is difficult to enter breast cancer cells through the cell membrane, thereby obtaining natural drug resistance [95]. In addition, an acidic environment can increase the expression level and transport activity of P-gp, and increase drug efflux, which is another important mechanism for TME to induce neoadjuvant chemotherapy drug resistance [96].

TME has special structural components that are different from the normal environment, and these structural components are closely related to the resistance of neoadjuvant chemotherapy drugs. For example, type 2 collagen secreted by tumor cell-related fibroblasts can reduce the absorption of neoadjuvant chemotherapy drugs and induce drug resistance [97]. At the same time, unsaturated fatty acids released by mesenchymal cells, a variety of chemokines secreted by neutrophils, glutathione sulfhydryl transferase Pi and many cytokines can all induce drug resistance in breast cancer cells [98,99,100].

TME fully mediates the drug resistance of breast cancer cells, which is a more difficult problem in the process of overcoming drug resistance. At present, studies have shown that for the HIF-1α signaling pathway induced by hypoxic environment, relevant HIF inhibitors can be used. Clinical trials have made progress in a variety of tumors, but they have not yet been used in neoadjuvant chemotherapy for breast cancer [101]. Aiming at the acidic environment of TME, a therapeutic strategy to target and inhibit the acidic microenvironment has also been developed, which has a good prospect in overcoming drug resistance [102,103].

### 2.9. Exosomes

Exosomes are membranous vesicles secreted after the fusion of intracellular multivesicular bodies and cell membranes [104], which can be released by a variety of normal cells and tumor cells. Exosomes contain a variety of biologically active molecules, such as proteins, lipids, nucleic acids, etc. [105]. Exosomes are mainly involved in the transmission of information between cells [106]. With the broadening of breast cancer research findings, it has been found that exosomes are also closely related to breast cancer cells [107], especially in mediating the resistance of breast cancer cells to neoadjuvant chemotherapy drugs.

Current studies have found that exosomes can transmit miRNA and proteins between breast cancer cells to produce drug resistance [108]. LV et al. [109] have shown that breast cancer cells that have developed drug resistance can transmit P-gp to other tumor cells through exosomes, thereby increasing the expression of P-gp in breast cancer cells that have not developed drug resistance, increasing drug efflux and producing drug resistance. In addition, exosomes can also transport MDR-1 and miRNAs to enable sensitive cells to acquire drug resistance [110,111]. Another mechanism by which exosomes mediate drug resistance is through avoiding the onset of neoadjuvant chemotherapy drugs through immune escape. Exosomes can carry the TGF-β of drug-resistant cells, deliver it to sensitive cells, change the cell’s response to IL-2, and escape the supervision of immune cells to produce drug resistance [112]. Exosomes can also enhance cell autophagy and inhibit cell apoptosis, and induce drug resistance [113].

In addition to the exosomes secreted by breast cancer cells themselves that can render sensitive cells drug-resistant, exosomes derived from other stromal cells in TME can also decrease the drug sensitivity of cells. Zheng et al. [114] found that miR-21 in exosomes secreted by M2 type macrophages can regulate signaling pathways, enhance the ability of cells to resist apoptosis, and inducing drug resistance. Boelens et al. [115] found that stromal cells and breast cancer cells interact through exosomes to activate nearby signals, and then develop drug resistance. Relevant studies have shown that exosomes secreted by mesenchymal stem cells (MSCs) and fibroblasts can induce drug resistance in breast cancer cells [116,117]. Exosomes can transmit information between tumors and between tumors and other cells. They are also a good carrier for targeted drug delivery. For resistance to neoadjuvant chemotherapy in breast cancer, tumor-targeted therapy aiming at decreasing exosome uptake may be a new option.

Figure 2 summarizes the drug resistance mechanism of neoadjuvant chemotherapy in breast cancer.

## 3. Solutions to the Reversal of Resistance to Neoadjuvant Chemotherapy Drugs in Breast Cancer

Neoadjuvant chemotherapy drug resistance is an important issue in clinical breast cancer treatment. At present, many studies are combining new drugs or new therapies with traditional neoadjuvant chemotherapy drugs, which can significantly reverse drug resistance caused by neoadjuvant chemotherapy drugs alone. The following chapter describes some drugs that have the potential to reverse neoadjuvant chemotherapy resistance. Although these drugs have not yet been clinically applied in breast cancer, it is believed that with research advances, these novel regimens can play an important role in the treatment of breast cancer.

### 3.1. Combined Use of Chemotherapeutic Drugs

According to the existing research on drug resistance mechanisms, many treatment options have brought hope to reverse the resistance of breast cancer to neoadjuvant chemotherapy. One of the more commonly used methods to reverse drug resistance is the combined use of multiple chemotherapy drugs.

On the one hand, the combined use of multiple drugs that work through different molecular mechanisms can greatly improve the problem of drug resistance caused by alterations in a single mechanism, thereby ensuring the efficacy of the drug. For example, in the clinical application of neoadjuvant chemotherapy drugs, anthracyclines that act on DNA and taxanes that act on proteins are used in combination, and drugs such as cyclophosphamide and epirubicin are used at the same time. These drugs act on different targets and can greatly improve the problem of drug resistance caused by a single mechanism, thereby ensuring the efficacy of the drug. Among neoadjuvant chemotherapy drugs for breast cancer, both trastuzumab and pertuzumab specifically act on the HER2 target, and their combined application can significantly enhance the sensitivity of breast cancer cells to neoadjuvant chemotherapy drugs. However, this method could inevitably enhance drug toxicity. In clinical treatment, it is also necessary to pay attention to the interaction between different drugs, detect the plasma drug concentration in time, and change the dose to obtain the best therapeutic effect.

On the other hand, while using neoadjuvant chemotherapy drugs, increasing the use of drugs that can block the drug resistance mechanism of tumors can restore the sensitivity of drug-resistant cells to drugs and enhance the efficacy of drugs. Most of these drugs are P-gp inhibitors, but due to high toxicity and pharmacokinetic effects, they have not been used in clinical practice [118]. The current research direction is to develop more efficient and low-toxic chemical drug reversal agents. Dual specificity phosphatase6 (DUSP6) inhibitors and histone deacetylase inhibitors (HDACI) have been identified as agents with the potential to reverse tumor drug resistance warranting follow-up research associated with drug development [119,120].

### 3.2. Chinese Medicine Reversal Agents

With the modern development of Chinese medicine, herbs and extractions from traditional Chinese medicine as drugs for treating tumors have gradually demonstrated their unique curative effects in the clinic [121,122,123,124]. More and more studies have shown that traditional Chinese medicine plays an important role in the prevention and treatment of breast cancer, contributing to the reversal of drug resistance of breast cancer. Moreover, compared with the above-mentioned chemical drug reversal agents, traditional Chinese medicine has the characteristics of safer, multi-component, multi-stage, and multi-targeted action. This makes traditional Chinese medicine monomers and extracts gradually attract more researchers’ attention as tumor drug reversal agents.

Many traditional Chinese medicine preparations have been developed and used in clinical practice, such as Elemene Injection and Shenqi Fuzheng Injection. The main active ingredient of Elemene Injection is a mixture of β-, γ-, and σ-elemene, which is an anti-cancer active ingredient extracted from *Curcuma wenyujin* Y.H.Chen. The combination of Elemene Injection and neoadjuvant chemotherapy drugs can reverse drug resistance of breast cancer cells, and its mechanism of action is mainly to inhibit the expression of P-gp, and reverse drug resistance of breast cancer cells through exosome and EMT inhibition [125,126,127,128]. Shenqi Fuzheng Injection is an injection made with *Codonopsis* and *Astragalus* as the main components. As early as 1999, Shenqi Fuzheng Injection has been formally approved by CFDA for adjuvant anti-tumor therapy [129]. Regarding its mechanism of reversing tumor drug resistance, the main mechanism is to induce cell cycle arrest and promote cell apoptosis [130]. However, it has not been used in neoadjuvant chemotherapy for breast cancer, and its mechanism of action should be further explored in the hope that it can be used in clinical practice in the future.

Table 1 shows some Chinese medicine reversal agents that act on breast cancer along with their mechanism of action. It can be seen that, in addition to the above-mentioned traditional Chinese medicine compound prescriptions, there are many monomer drugs extracted from traditional Chinese medicines that have been proven to have the effect of reversing tumor resistance. Honokiol and magnolol, which are extracted from the plant *Magnolia officinalis* Rehd. et Wils., have a variety of pharmacological effects, such as inducing long-lasting central muscle relaxation and central nervous system inhibition, as well as anti-inflammatory, antibacterial, anti-ulcer, anti-tumor, etc. Hyo-Kyung Han et al. [131] evaluated the inhibitory effects of honokiol and magnolol on P-pg activity, and found that honokiol can inhibit P-gp activity through a competitive mechanism, and both compounds can inhibit the expression of P-gp and help to improve the resistance of breast cancer to neoadjuvant chemotherapy drugs. Curcumin is derived from *Curcuma longa* L. In related studies, the combined treatment of curcumin and the neoadjuvant drug doxorubicin used in breast cancer can reduce the excessive efflux of doxorubicin caused by the overexpression of ABCB4, thereby enhancing the efficacy of doxorubicin [132]. Saikosaponin D is derived from *Bupleurum chinense* DC. It has been shown to have anti-inflammatory, antibacterial and anti-tumor effects. Relevant studies have shown that Saikosaponin D can enhance the sensitivity of breast cancer multidrug resistant cells MCF-7/ADR cells to chemotherapeutic drugs by reducing the expression of MDR1 and P-pg, which may effectively reverse the effects of neoadjuvant chemotherapy drug resistance [133]. Berberine is an alkaloid, extracted and purified from *Coptis chinensis* Franch. It has a variety of pharmacological effects, such as antibacterial, antihypertensive, antiarrhythmic and antitumor [134]. For reversing the drug resistance of breast cancer cells, different doses of berberine mediate different mechanisms. When using a low dose (5 µM) ofberberine, the AMPK/HIF-1α signaling pathway is inhibited and the expression of P-gp decreases, thereby increasing the sensitivity of breast cancer cells to chemotherapy drugs. The high dose (40 µM) of berberine can regulate the AMPK/HIF-1α signaling pathway, which in turn activates the expression of p53 and directly induces breast cancer cell apoptosis [135]. Resistance to paclitaxel drugs is a major obstacle in neoadjuvant chemotherapy for breast cancer, and studies have shown that gambogic acid can enhance the sensitivity of breast cancer cells to paclitaxel [136]. Gambogic acid is derived from *Garcinia hanburyi* Hook. f., it has anti-proliferative effects in triple-negative breast cancer cells [137]. It can also induce breast cancer cell apoptosis by inhibiting SHH signaling pathway, and has potential to be used as a combination drug in the neoadjuvant treatment of breast cancer. Ligustrazine is derived from the root of *Ligusticum chuanxiong* Hort. Various studies have shown that ligustrazine can block the G0/G1 phase of the cell cycle, thereby inhibiting DNA synthesis, and inducing breast cancer cell apoptosis, thereby reversing breast cancer cell resistance to neoadjuvant chemotherapy drugs [138].

Due to the complex composition of traditional Chinese medicine agents, numerous targets, and unclear mechanism of action, its follow-up research should also conduct efficacy and safety evaluations to accelerate the transformation to the clinic.

### 3.3. Gene Modifications

In recent years, the development of genetic engineering technology has made it possible to reverse the resistance of neoadjuvant chemotherapy drugs at the genetic level. Among them, nucleic acid-based technologies, such as siRNA, antisense oligonucleotide (ASO), mRNA, etc., have shown great potential in regulating the expression of tumor-related genes [140].

Mutations in the MEN1 gene encoding menin can cause tumors in multiple endocrine organs. For example, the occurrence and development of breast cancer are closely related to menin. Current studies have shown that inhibitors of the menin/MLL1 complex are effective in some cancers, but they are not effective in the treatment of breast cancer [141]. Recent studies have found that ASO targeting menin mRNA has a greater advantage than siRNA, and has a better curative effect in the treatment of triple-negative breast cancer [142]. More importantly, in vitro menin silencing can have a synergistic effect with the taxane drug docetaxel in neoadjuvant chemotherapy. Moreover, menin-ASO can be used in combination with neoadjuvant chemotherapy drugs that induce DNA damage or PARP inhibitors that inhibit DNA damage. It can promote cell apoptosis, which provides new ideas for neoadjuvant chemotherapy for breast cancer.

The upregulation of some pro-apoptotic genes, such as p53, can also reverse resistance of neoadjuvant chemotherapy drugs. In breast cancer cells, the tumor suppressor gene p53 is often mutated to cause abnormal expression, thereby activating the promoter of the MDR1 gene and increasing its expression. At present, the p53 gene using adenovirus as a vector, and the antisense gene of the mutant p53 gene can be used to introduce the wild-type p53 gene can be potentially used to reverse the resistance to neoadjuvant chemotherapy drugs in breast cancer [143,144].

MicroRNA (miRNA) is an endogenous 18–23-nucleotide small noncoding RNA, which can bind to the 3′-untranslated region (3′-UTR) of specific target messenger RNAs (mRNAs). Abnormal expression of miRNA, such as the oncomiRNA, microRNA-21, is overexpressed in breast cancer, which may lead to the occurrence of breast cancer, thus the regulation of miRNA is also a potential target for the treatment of breast cancer [145]. Anti-microRNA oligonucleotides (AMOs) are synthetic oligonucleotides that can complementally bind to the corresponding target miRNA, thereby inhibiting the expression of miRNA [146]. At present, most AMOs are designed for high-expressed miRNAs in breast cancer cells. Recent studies have shown that inhibiting low-expressed miRNAs in breast cancer cells, such as mir-148a of the mir-148/152 family, can also effectively inhibit the proliferation of breast cancer cells to treat breast cancer [147]. These studies show that the combination of AMO for the suppression of related miRNA expression and neoadjuvant chemotherapy has great potential to enhance the efficacy of neoadjuvant chemotherapy drugs and bring new hope for the treatment of breast cancer.

Genetic engineering mainly blocks the expression of drug-resistant genes, or acts synergistically with neoadjuvant chemotherapy drugs to promote breast cancer cell apoptosis. The development and clinical application of genetic engineering technology can effectively reverse the resistance of neoadjuvant chemotherapy drugs and improve the effect of clinical treatment of breast cancer. However, the drug resistance mechanism of breast cancer is complex, including multiple pathways and factors. For different mechanisms, the research direction should be focused on more approaches and more targets to achieve more therapeutic effects.

### 3.4. Immune Regulation

The current use of the immune system to reverse resistance to neoadjuvant chemotherapy drugs is mainly associated with the application of antibodies and the application of many cytokines in the body’s own immune system.

Antibodies include mainly P-gp antibodies and APO-1 monoclonal antibodies. P-gp antibodies can recognize the epitope of the P-gp membrane, competitively inhibit the pumping function of P-gp, reduce neoadjuvant chemotherapy drug efflux, enable drug accumulation in the cell, and reverse resistance to neoadjuvant chemotherapy drugs [148]. In addition, antibodies can mediate immune responses and participate in the reversal of drug resistance. The APO-1 monoclonal antibody is an antibody against the glycoprotein FAS on the cell membrane surface, which can bind to APO-1 on the breast cancer cell membrane surface to induce cell apoptosis [149]. For some breast cancer cells with low FAS expression, the FAS antigen expression vector can also be transfected into cells to promote high expression before using APO-1 monoclonal antibodies.

The cytokines of the body’s immune system, such as TNF-α, INF-α and IL-2, can reduce the expression levels of MDR1 gene mRNA and P-gp, increase the sensitivity of breast cancer cells to neoadjuvant chemotherapy drugs, and reverse resistance to neoadjuvant chemotherapy drugs [150].

### 3.5. Changing the Tumor Microenvironment

TME and tumor cells interact with each other. Tumor cells can create a suitable living environment for themselves by changing TME, and TME in turn regulates tumor cells, including promoting drug resistance. At present, TME is a hot spot in tumor research. It can provide new ideas for tumor treatment and reverse tumor drug resistance to a certain extent [151].

In addition to the above-mentioned HIF inhibitors and targeted inhibitors developed for the TME hypoxic environment and acidic environment, a current research focus is the use of nanomaterials when analyzing TME models, and the focus on the mechanism of nanomedicine targeting CAFs [152].

Moreover, the combination of anti-angiogenic therapy and immunotherapy can lead to a reversal of the immunospressive TME [153]. In breast cancer, anti-angiogenic therapy can induce the normalization of tumor blood vessels, and promote the recruitment of immune cells and the maturation of dendritic cells (DC) [154,155]. The use of immune checkpoint inhibitors can further alleviate the immunosuppressive state and promote the normalization of TME, thereby reversing resistance of neoadjuvant chemotherapy drugs induced by TME.

More importantly, the heterogeneity of TME is related to breast cancer subtypes, and the treatment plan for TME will help realize the potential of “precision medicine”.

Figure 3 summarizes the solutions to the reversal of resistance to neoadjuvant chemotherapy drugs in breast cancer.

## 4. Conclusions

With the rapid development of molecular biology, breast cancer treatment has made great progress. Among these treatments, neoadjuvant chemotherapy is suitable for patients with mid-stage and locally advanced breast cancer, and plays a very important role in the treatment of breast cancer. Through the development of individualized treatment plans, the sensitivity of patients toward neoadjuvant chemotherapy can increase, thus improving survival rates. At present, one of the biggest obstacles to the clinical application of neoadjuvant chemotherapy in the treatment of breast cancer is the development of drug resistance. It can cause patients to be insensitive to neoadjuvant chemotherapy drugs, decreasing treatment effects, and even increasing the progression of disease, delaying the time for radical treatment of the tumor. Therefore, reversing the drug resistance of tumors is a necessary task for the treatment of breast cancer.

As mentioned above, the mechanism of breast cancer drug resistance is more complicated, and it is a multi-factorial and multi-step process. It is the result of interactions of cancer cells within the tumor, as well as closely related to the role of breast cancer cells and their surrounding environment. Due to the complexity of resistance of neoadjuvant chemotherapy drugs, fully understanding its mechanism is still an important challenge today. With the continuous development of pharmacology, new mechanisms of breast cancer drug resistance have emerged, and at the same time, novel methods of reversing resistance to neoadjuvant chemotherapy drugs have been gradually used in clinical treatment.

At present, most of the neoadjuvant chemotherapy options for breast cancer are still using a combination therapy program, which can not only increase the efficacy of the drug, but also effectively prevent the decrease in efficacy due to tumor resistance. Presently, many chemotherapeutic drugs or traditional Chinese medicine reversal agents have been developed, and they have shown good curative effects in clinical applications. Meanwhile, the rise of genetic engineering has attracted a large number of researchers to examine the reversal of tumor resistance at the genetic level. For example, the combination of RNAi technology and gene therapy vectors has shown good clinical application prospects. In addition, the research progress on the microenvironment surrounding tumor cells provides a new treatment plan for reversing tumor drug resistance. The heterogeneity of the TME is related to breast cancer subtypes, which indicates that the treatment plan for TME will help to achieve individualized treatment plans in the future.

It is believed that with the development of novel medical concepts and technologies, the resistance of neoadjuvant chemotherapy drugs can be reversed, and neoadjuvant chemotherapy can increase its efficacy in the treatment of breast cancer and improve the clinical treatment effect.

## Figures and Tables

**Figure 1 ijms-22-09644-f001:**
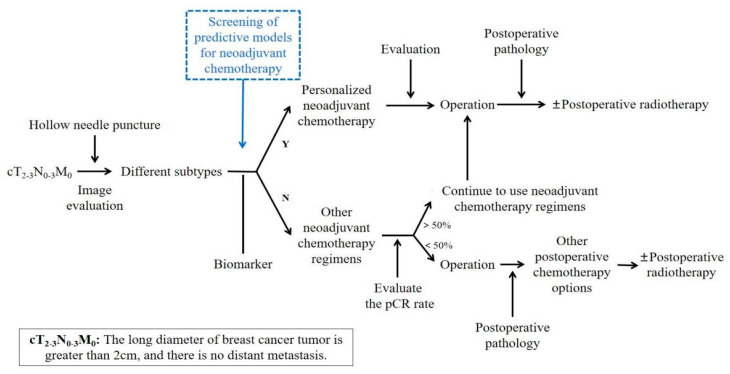
Personalized neoadjuvant chemotherapy.

**Figure 2 ijms-22-09644-f002:**
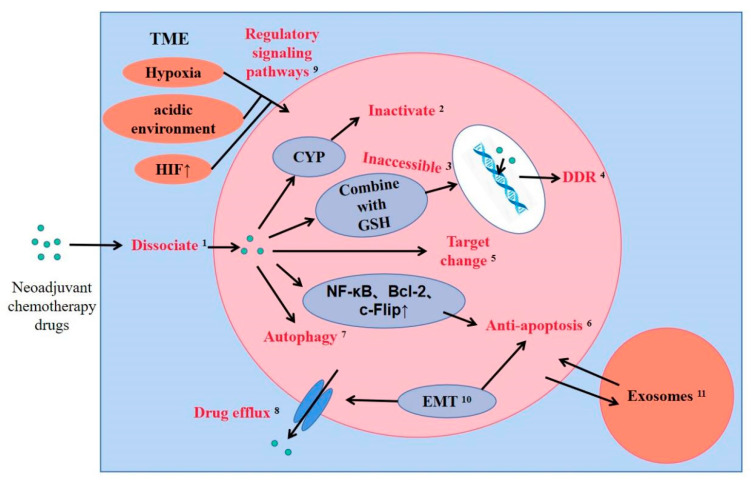
Drug resistance mechanism of neoadjuvant chemotherapy in breast cancer. ^1^ Weakly basic drugs dissociate in acidic TME. ^2^ CYP can inactivate drugs. ^3^ Platinum cannot enter the nucleus to play a role after being combined with GSH. ^4^ Drugs that directly or indirectly damage DNA can induce breast cancer cells to produce DDR. ^5^ Trastuzumab can induce nuclear HER4 up-regulation and nuclear HER2 translocation. ^6^ Over-expression of NF-κB, Bcl-2 and c-Flip can inhibit cell apoptosis. ^7^ Endoplasmic reticulum stress or DNA damage can induce autophagy. ^8^ The ABC transporter family mediates drug efflux, mainly P-gp, MRP and BCRP. ^9^ TME can induce drug resistance through a variety of signaling pathways. ^10^ The overexpression of N-Cadherin and Vimentin in EMT can inhibit cell apoptosis and promote drug efflux. ^11^ Exosomes can transmit miRNA and protein between cells to produce drug resistance.

**Figure 3 ijms-22-09644-f003:**
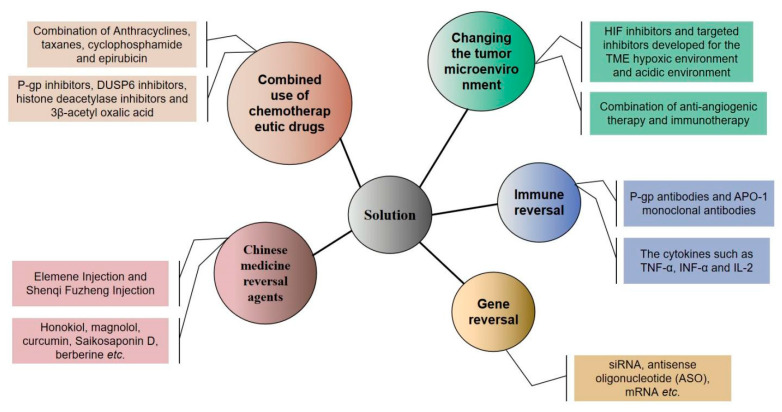
Strategies to combat resistance to neoadjuvant chemotherapy drugs in breast cancer.

**Table 1 ijms-22-09644-t001:** Chinese medicine reversal agents for breast cancer and their mechanism of action.

Compound	Molecular Formula	Resource	Potential Targets	Function Study	Renference
Honokiol, magnolol	C_18_H_18_O_2_	*Magnolia officinalis* Rehd. et Wils.	↓P-gp	In vitro	[131]
Curcumin	C_21_H_20_O_6_	*Curcuma longa* L.	↓ABCB1, ABCG2 and ABCCs	In vitro	[132]
Saikosaponin D	C_42_H_68_O_13_	*Bupleurum chinense* DC.	↓MDR1 and P-gp	In vitro	[133]
Berberine	C_20_H_18_NO_4_^+^	*Coptis chinensis* Franch.	Low dose: ↓AMPK-HIF-1-P-gp pathway High dose: ↓AMPK-HIF-1α-p53 pathway and ↑apoptosis	In vitro and in vivo	[135]
Gambogic acid	C_38_H_44_O_8_	*Garcinia hanburyi* Hook. f.	↓SHH signaling pathway	In vitro and in vivo	[136]
Ligustrazine	C_8_H_12_N_2_	*Ligusticum chuanxiong* Hort.	↑apoptosis	In vitro and in vivo	[138]
Paris saponinⅦ	C_51_H_82_O_21_	*Trillium tschonoskii* Maxim.	↓MDR1 and P-gp	In vitro and in vivo	[139]

↓—low expression; ↑—high expression.

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
