# Peer review of "New Advances in the Research of Resistance to Neoadjuvant Chemotherapy in Breast Cancer"

_ijms, 2021, doi:10.3390/ijms22179644_

Round 1

Reviewer 1 Report

The authors reviewed the advances in drug resistance studies focusing on breast cancer-directed neoadjuvant chemotherapy. This is a topic worthy to be reviewed for understanding the current problems and developing future strategies. However, the current manuscript is not ready to be published in IJMS.

Major

  1. Neoadjuvant chemotherapy: Although the focus of this review is on neoadjuvant chemotherapy, the descriptions are little to do with neoadjuvant chemotherapy. It is strongly recommended to describe it including examples.
  2. Illustrations: One figure on autophagy is included, but autophagy is not a representative topic in this review. Also, the figure is not fully explained. It is strongly recommended to have a few key illustrations that represent the major issues in neoadjuvant chemotherapy.
  3. Linkage to breast cancer: Linkage to breast cancer is insufficient. The subjects in the current manuscript are mostly applicable to all solid tumors.
  4. Combinatorial application: More detailed descriptions with examples are highly recommended.
  5. Chinese medicine: This is a new insight and potentially interesting. However, the current description is superficial and data are almost all in vitro. Also, molecular formulas are not informative and the potential targets are poorly presented.

Minor

  1. line 292: TME is described to refer to the area but TME is more than the area.
  2. line 305: The phrase “In TME, the overexpression of HIF-1a can …” is ambiguous. The question is whether HIF-1a is overexpressed by tumor cells or any other cells.
  3. line 399: Please explain “element injection” and “Shenqi Fuzheng injection.”

Author Response

-----

Point 1: Neoadjuvant chemotherapy: Although the focus of this review is on neoadjuvant chemotherapy, the descriptions are little to do with neoadjuvant chemotherapy. It is strongly recommended to describe it including examples.

 Response 1: Thanks for the reviewer’s comments, which are very helpful for the improvement of this review. In response to the insufficient description of neoadjuvant chemotherapy, we have already described the neoadjuvant chemotherapy for breast cancer in more detail in the background section of the article (line 47-68). Besides, subsequent methods to reverse tumor drug resistance are based on the example of neoadjuvant chemotherapy drugs in breast cancer.

Point 2: Illustrations: One figure on autophagy is included, but autophagy is not a representative topic in this review. Also, the figure is not fully explained. It is strongly recommended to have a few key illustrations that represent the major issues in neoadjuvant chemotherapy.

Response 2: Thanks for the reviewer’s important suggestion. At your suggestion, we have deleted the illustration about autophagy, and added an illustration as Figure 1 representing the new treatment plan of neoadjuvant chemotherapy for breast cancer.

Point 3: Linkage to breast cancer: Linkage to breast cancer is insufficient. The subjects in the current manuscript are mostly applicable to all solid tumors.

Response 3: Thanks for the reviewer’s valuable comments, which are very helpful for the improvement of this review. Regarding the link between this article and breast cancer, we mainly summarize and discuss neoadjuvant chemotherapy for breast cancer. The examples of various mechanisms in the article are derived from breast cancer. Because some mechanisms are common to breast cancer and all solid tumors, there are many mechanisms in this article that apply to all solid tumors. But there are also some mechanisms in the article, such as the drug pumping effect of BCRP and the change of drug target of trastuzumab, which are are unique to breast cancer. Moreover, many of the treatments to reverse drug resistance mentioned in the article are also unique to breast cancer.

Point 4: Combinatorial application: More detailed descriptions with examples are highly recommended.

 Response 4: Thanks for the reviewer’s suggestion. We have added some examples of the combined application of neoadjuvant chemotherapy drugs for breast cancer (line 417-425).

Point 5: Chinese medicine: This is a new insight and potentially interesting. However, the current description is superficial and data are almost all in vitro. Also, molecular formulas are not informative and the potential targets are poorly presented.

 Response 5: Thanks to the reviewers for their valuable comments on this section. We have described the relevant data of each Chinese medicine monomer and its mechanism in more detail, and many monomers also have in vivo experimental data (line 468-502). Hope these supplements explain the content of the table in detail.

 Minor

Point 1: line 292-299: TME is described to refer to the area but TME is more than the area.

 Response 1: Thanks to the reviewers for their valuable comments. The concept of TME has been corrected in line 323.

Point 2: line 305-312: The phrase “In TME, the overexpression of HIF-1a can …” is ambiguous. The question is whether HIF-1a is overexpressed by tumor cells or any other cells.

 Response 2: Thanks for the reviewer’s important suggestion. We are so sorry about the mistakes. Reference [90] has mentioned that HIF-1a is overexpressed in TME, here is our improper expression, which has been corrected (line 335-337).

Point 3: line 399-406: Please explain “element injection” and “Shenqi Fuzheng injection.”

 Response 3: Thanks to the reviewers for their important comments. Elemene Injection and Shenqi Fuzheng Injection are both injections made with a variety of traditional Chinese medicines as the main active ingredients. We have added a description of their composition in the article (line 450-452, 456-457).

Reviewer 2 Report

Interesting paper on drug resistance emerging during breast cancer treatment. However, the authors mainly focused on the importance of traditional chinese medicine in countering drug resistance. This is somewhat understandable and provides information to explain the role of traditional chinese medicine in the modern approach to breast cancer treatment.
I suggest modifying the title of the article so that the potential reader can anticipate the content of the paper.
I am also afraid that a reader from the area of European countries may be surprised by the content of the paper. It should also be remembered that traditional chinese medicine is available mainly in Asian countries and can be treated as a regional curiosity by readers from Europe or the USA.
Nevertheless, I find the article very interesting and provides an overview of the causes of drug resistance in breast cancer.

Author Response

Thanks for the reviewer's comments, which are very helpful for the improvement of this review. According to your suggestion, we have revised the title of this article to "Mechanism and solution of resistance to neoadjuvant chemotherapy drugs in breast cancer", which can represent and summarize the overall content of the article. For the description of traditional Chinese medicine in the article, we have added the relevant background of Chinese medicine so that relevant readers can better understand it. In addition, we have perfected the relationship between this article and neoadjuvant chemotherapy drugs for breast cancer. At the same time, we expound the mechanism of resistance to neoadjuvant chemotherapy drugs in breast cancer from more aspects and provide more options for reversing drug resistance.

Round 2

Reviewer 1 Report

The authors mostly responded to the comments to the original manuscript, but there are a few issues to be further revised.

  1. Figure 1:  Some phrases in the diagram are not clear. For instance, "continue 4-6" does not explain the intended chemotherapy.  Figure legend should be provided and explain the terms such as cT2-3N0-3M0, >50% (not clear what the evaluation is), etc.
  2. Mechanism and solution: The title of this article is "mechanism and solution," but this part is not fully described. It is recommended to add a figure that summarizes the mechanism and solution, with a comprehensive figure legend.

Author Response

Point 1: Figure 1:  Some phrases in the diagram are not clear. For instance, "continue 4-6" does not explain the intended chemotherapy.  Figure legend should be provided and explain the terms such as cT2-3N0-3M0, >50% (not clear what the evaluation is), etc.

Response 1: Thanks for the reviewer’s comments, which are very helpful for the improvement of this review. We have solved the problem of ambiguity raised by the reviewer in Figure 1.

Point 2: Mechanism and solution: The title of this article is "mechanism and solution," but this part is not fully described. It is recommended to add a figure that summarizes the mechanism and solution, with a comprehensive figure legend.

Response 2: Thanks for the reviewer’s important suggestion. At your suggestion, we have added Figure 2 and Figure 3 to summarize the mechanism and solution.
